# Hydroxyapatite Affects the Physicochemical Properties of Contemporary One-Step Self-Etch Adhesives

**DOI:** 10.3390/ma15228255

**Published:** 2022-11-21

**Authors:** Yutaro Motoyama, Monica Yamauti, Masatoshi Nakajima, Masaomi Ikeda, Junji Tagami, Yasushi Shimada, Keiichi Hosaka

**Affiliations:** 1Department of Cariology and Operative Dentistry, Graduate School of Medical and Dental Sciences, Tokyo Medical and Dental University, 1-5-45 Yushima, Bunkyo-ku, Tokyo 113-8510, Japan; 2Department of Restorative Dentistry, Graduate School of Dental Medicine, Hokkaido University, Kita 13 Nishi 7, Sapporo 060-8586, Japan; 3Department of Regenerative Dental Medicine, Graduate School of Medical and Dental Sciences, Tokushima University, 3-18-15 Kuramoto-Cho, Tokushima 770-8504, Japan; 4Oral Prosthetic Engineering, Graduate School of Medical and Dental Sciences, Tokyo Medical and Dental University, 1-5-45 Yushima, Bunkyo-ku, Tokyo 113-8510, Japan

**Keywords:** one-step self-etch adhesive system, water sorption, flexural strength, modulus of elasticity, pH, hydroxyapatite

## Abstract

The study aimed to evaluate the influence of the manipulation surfaces on the physical properties of one-step self-etch adhesives (1-SEAs). Scotchbond Universal (SBU), Clearfil Universal Bond Quick ER (UBQ), and an experimental adhesive (UBQ_exp_) were manipulated on different surfaces: manufacturer’s Teflon-based dispensing dish (TD) or hydroxyapatite plate (HA). After manipulation of the adhesives, the pH of each 1-SEA was measured. Samples of each adhesive/manipulation surface were prepared and subjected to water sorption (WS)/solubility (SL) and flexural strength tests. The modulus of elasticity (E) was measured in dry and wet conditions before and after 24 h water storage, and the percentage of variation of E (ΔE) was calculated. Results were analyzed using the *t*-test with Bonferroni corrections (α = 0.05). When adhesives were manipulated on the HA plate, there was a significant increase in the adhesives’ pH. WS and SL of all 1-SEAs decreased when the HA was used. Only SBU showed higher flexural strength when manipulated on the HA compared to the manipulation on TD under dry and wet conditions. For each 1-SEA, the use of HA resulted in significantly higher E in dry and wet conditions. ΔE of all adhesives was smaller with the manipulation on HA than on TD. It was concluded that the manipulation of 1-SEA on a hydroxyapatite plate considerably affected the adhesives’ properties.

## 1. Introduction

Regardless of the fillings’ dimensions, direct composite restorations have been widely advocated as efficacious minimally invasive treatment options [1,2]. The clinical performance and durability of direct composite restorations depend on the properties of used adhesive systems [3], among other factors. In recent decades, adhesive systems have been optimized and simplified, aiming for a less technique-sensitive bonding procedure [3,4,5]. The use of one-step self-etch adhesive systems (1-SEAs) has surpassed conventional multi-step methods. 

However, some concerns related to the bonding durability of 1-SEAs have arisen. The bonding durability may diminish over time because of adhesives’ high hydrophilicity, mainly attributed to their chemical composition. The polar acidic functional monomers, hydrophilic monomers (i.e., 2-hydroxyethyl methacrylate—HEMA), and water increase 1-SEAs’ hydrophilicity, favoring water absorption [6,7], thus significantly reducing their elastic modulus and tensile strength [7,8]. The presence of HEMA and the acidity of 1-SEAs may also affect their polymerization [9]. The non-evaporated and residual solvents could inhibit the polymerization of the resin monomers and cause the elution of unpolymerized monomers, thereby increasing the amount of water sorption by forming water immersion spaces [10] and jeopardizing the long-term prognosis of bonded restoration. Clinically, direct composite restorations bonded with 1-SEAs underperformed those bonded with 2-SEAs in terms of discoloration, marginal discoloration, fracture, biofilm accumulation, postoperative hypersensitivity, and overall satisfaction [11].

The ISO 4049-2019 standard refers to the requirements for dental polymer-based restorative materials, and some modifications can be made to standardize the screening evaluation for testing dental adhesives [7,12]. Nevertheless, the physical properties are examined indirectly and under controlled experimental conditions that do not reproduce the oral conditions or the tooth surfaces. For instance, 1-SEAs are manipulated and polymerized after the evaporation of solvents on an inert container. Consequently, the effect of the enamel and dentin substrates, rich in inorganic content, especially hydroxyapatite, is not considered. Additionally, 1-SEAs absorb water that could affect the hybrid layer formation and imperil the longstanding durability of the resin–tooth bond [13]. The experimental laboratory conditions should consider some in situ clinical conditions when analyzing the properties of adhesive systems, such as humidity.

The chemical reaction between the ionic monomer with acidic phosphate or carboxylic functional groups and hydroxyapatite significantly affects the bond durability of SEAs [14]. The acidic monomers partially remove the smear layer formed on the prepared tooth surface and chemically bond with the remaining hydroxyapatite, increasing the initial bond strength to dentin. In particular, the functional monomer in 1-SEAs reacts with hydroxyapatite, originating a calcium salt of the monomer at the adhesive interface and reducing the number of polar groups. After that chemical reaction, there might be a change in the acidity and hydrophilicity of 1-SEAs, improving the polymerization of 1-SEAs and their physical properties.

Therefore, the goal was to investigate if the manipulation surfaces of 1-SEAs (manufacturers’ Teflon-based dispensing dish and hydroxyapatite plate) would interfere with the materials’ pH, water sorption and solubility, flexural strength, and the modulus of elasticity of their polymers. The null hypothesis tested was that applying 1-SEAs on hydroxyapatite does not affect their pH values, water sorption/solubility, flexural strength, and modulus of elasticity.

## 2. Materials and Methods

### 2.1. Manipulation and Preparation of the Adhesives

Two commercially available 1-SEAs, Scotchbond Universal (SBU; 3M ESPE; St. Paul, MN, USA), Clearfil Universal Bond Quick (UBQ; Kuraray Noritake Dental Inc.; Tokyo, Japan), and one experimental 1-SEA (UBQ_exp_; Kuraray Noritake Dental Inc.; Tokyo, Japan) were used in this study. Their compositions and the manufacturer’s instructions are shown in Table 1. 

Figure 1 illustrates the manipulation of the 1-SEAs. Hydroxyapatite (Ca_10_(PO_4_)_6_(OH)_2_; 100%) plates (30 mm in diameter, 2 mm thick) were manufactured (HOYA Technosurgical Corporation, Tokyo, Japan). They were smoothened using 600-grit SiC papers under running water.

Each 1-SEA was dispensed on the manufacturers’ Teflon-based dispensing dish (TD) or the hydroxyapatite plate (HA) and agitated with a micro brush for 2 min. The solvents of each adhesive were carefully evaporated using a dental air syringe for 10 min at 15 cm (pressure of 3.8 kgf/cm^2^) in the dark until the weight was stable.

### 2.2. Measurement of Adhesives’ pH

Each 1-SEA was diluted (1:5), and its pH was measured in darkness to avoid a spontaneous polymerization reaction. For the dilution, 200 μL of each 1-SEA manipulated as described above was collected, added to 1000 μL of distilled water and ethanol, and stirred well until the solution was utterly non-transparent, with a milky aspect. Subsequently, the diluted adhesive solutions were transferred to a microtube and centrifuged for 10 min. As a result of centrifugation, adhesive solutions were separated into two phases. The pH measurement was performed only on the supernatant using a compact pH meter (Horiba ltd., Tokyo, Japan).

### 2.3. Water Sorption (WS) and Solubility (SL)

After the manipulation (item 2.1.), each adhesive system was dispensed into metal molds (6.0 mm diameter × 1.0 mm thick) to obtain disk-shaped specimens. The adhesive content in the mold was covered with a transparent thin polyester strip to avoid the effect of environmental oxygen on the monomers’ reaction. The adhesives in these molds were light-activated for 20 s with an LED light-curing unit (Pencure2000, Morita, light irradiance > 800 mW/cm^2^). The specimens were removed from the mold, turned over, and light-cured for additional 20 s to ensure adequate light-curing considering the bulk volume of 1-SEAs. After polymerization, the excess adhesive around the mold was carefully removed using a scalpel. The thickness and diameter of the specimens were measured using a digital caliper (ABS Solar Digimatic Caliper, Mitutoyo, Kanagawa, Japan), rounded to the nearest 0.01 mm, and these measurements were used to calculate the volume (V) of each specimen (in mm^3^). Water sorption and solubility were determined according to the ISO specification 4049-2019 standard, except for the specimens’ dimensions and period of water immersion (*n* = 7).

Subsequently, the specimens were placed in a desiccator loaded with anhydrous calcium sulfate (CaSO_4_). The samples were repeatedly weighed after 24 h intervals until a constant mass (M0) was obtained. Then, they were individually immersed in distilled water at 37 °C. After 24 h, the resin disks were gently dried with absorbent paper, weighed, and stored in distilled water until a constant mass (M1) was obtained. Then, the resin disks were stored in a dry state and weighed daily until a constant dry mass (M2) was obtained. Water sorption (WS) and Solubility (SL) were calculated using the following formula:$$\mathrm{WS}=\frac{\left(M1-M2\right)}{V}\;\mathrm{and}\;\mathrm{SL}=\frac{\left(M0-M2\right)}{V}$$
where M1 is the wet constant mass (μg) after water storage, M2 is the constant dry mass after the second desiccation, and V is the specimen volume in mm^3^.

### 2.4. Three-Point Flexural Bending Test

After manipulation, each 1-SEA was poured into beam-shaped silicon molds (1.0 × 1.0 × 10.0 mm^3^) and covered by a transparent thin polyester strip (Hawe Striproll; KerrHawe; Bioggio, Switzerland). The adhesives in these molds were light-activated for 20 s with an LED light-curing unit (Pencure2000, Morita, Tokyo, Japan, light output > 800 mW/cm^2^). After light curing, all the specimens were dried for 24 h in a desiccator. After the 24 h storage, half of the adhesives’ beam-shaped specimens (*n* = 8) were kept dry for 24 h (dry condition). The remaining specimens (*n* = 8) were immersed in distilled water for 24 h (wet condition). All beam-shaped specimens were subjected to a three-point flexural bending test to measure the modulus of elasticity (E). The three-point flexural bending test was performed with a miniature three-point bending stainless steel device (3-point bending jig, Ikegami Seiki Co., Ltd., Mahwah, NJ, USA) consisting of a supporting base with a 5 mm span and a loading piston. Three-point flexure was measured by centrally loading the adhesives’ specimens using a tabletop testing machine (EZ-SX, Shimadzu, Tokyo, Japan) and a displacement rate of 1 mm/min, sufficient to induce a 3% strain. Load–displacement values were converted to stress and strain. The modulus of elasticity (E) was calculated, in MPa, as the slope of the linear portion of the stress–strain curve from the following formula:$$E=\frac{{\mathrm{FL}}^{3}}{{4\mathrm{Dbh}}^{3}}$$
where F is the force (N), L is the span length (5.0 mm), D is the vertical deflection (mm) of the specimen, b is the width of test specimens (1.0 mm ± 0.1 mm), and h is the thickness (1 mm). 

The strain (ε) produced a three-point bending, which was calculated as
$$\varepsilon =\frac{6\mathrm{hd}}{L}$$
where h is the thickness of the beam (mm), d is the displacement of the beam (mm), L is the span length of the beam between the supports (5.0 mm), and ε is strain %.

Based on the data of the modulus elasticity obtained in wet and dry conditions, the variation of E between those conditions was calculated as ΔE (%).

### 2.5. Scanning Electron Microscope/Energy Dispersive X-ray Spectrometry Analysis

Six disk-shaped specimens from each adhesive/manipulation condition were prepared, light-cured for 20 s, mounted on brass stubs, and desiccated for 24 h. After sputter-coating the specimens with gold, their surfaces were observed using a scanning electron microscope (Feg-SEM; JSM-6701F, JEOL, Tokyo, Japan) at 1000× magnification. In addition, EDS analysis was performed with field-emission-gun SEM (Feg-SEM; JSM-6701F, JEOL, Tokyo, Japan) at 15 kV with an annular semiconductor detector for the calcium (Ca) at the 1-SEAs surface. The observed point was randomly selected for each material.

### 2.6. Statistical Analysis

In each test, the number of specimens per group (*n*) was calculated using the sample size determination method for two-tailed *t*-tests as follows: $$n=2\times \frac{{\left(1.96+0.84\right)}^{2}\times {\left(Standard\;Deviation\right)}^{2}}{\left(Mean\;Difference\right)}$$

The confidence level was set to 95%, and the statistical power level to 90%. The distribution of data and variance of data were analyzed by the Shapiro–Wilk test and Levene’s test. Data distributions indicated normality according to the Shapiro–Wilk test in all experimental groups (*p* > 0.05). However, the data variance was not equal among the groups (*p* < 0.05). Therefore, pH, water sorption/solubility, flexural strength, and modulus of elasticity data were analyzed using the *t*-test (Welch method) with Bonferroni corrections to examine statistically significant differences between the adhesives manipulated in the TD and HA. The *t*-test with Bonferroni correction was used for multiple comparisons at the 95% confidence level. Regression analyses were used to determine the correlations between both percent decrease of moduli of elasticity vs. water sorption. The statistical analyses were performed using the SPSS v27.0 software (IBM, Armonk, NY, USA).

## 3. Results

### 3.1. pH 

The pH values of all three 1-SEAs are compiled in Table 2. There was no statistically significant difference in the pH between the 1-SEAs manipulated in TD. On the other hand, when the 1-SEAs were manipulated on the HA, their pH significantly increased (*p* < 0.05).

### 3.2. Water Sorption (WS) and Solubility (SL)

The WS and SL values of all three adhesives are outlined in Table 3. The *t*-test with Bonferroni corrections revealed that the HA affected WS and SL (*p* < 0.05). The WS values of all adhesives manipulated on the HA were significantly lower compared to the values obtained when adhesives were manipulated on the TD (Figure 2). Except for SBU (*p* > 0.05), the use of HA lowered the SL (*p* < 0.05). When comparing materials for both manipulating surfaces, the WS values were in the order of UBQ_exp_ > SBU > UBQ. 

The SL values were UBQ_exp_ > UBQ = SBU for the manipulation on TD and UBQ_exp_ > SBU > UBQ for the manipulation on HA.

### 3.3. Three-Point Flexural Bending Test

The results of the flexural strength and modulus of elasticity of 1-SEAs manipulated on different surfaces and under distinct storage conditions are shown in Table 4 and Table 5 and Figure 3. The flexural strength of all 1-SEAs presented higher values under dry storage conditions when manipulated on both surfaces. Only SBU showed increased flexural strength when manipulated on HA compared to TD (*p* < 0.05).

E significantly increased when all 1-SEAs were manipulated on HA for both storage conditions (*p* < 0.05). The wet storage condition reduced the modulus of elasticity of all 1-SEAs when they were manipulated on TD and HA (*p* < 0.05). The ΔE was slightly lower when the adhesives were manipulated on HA.

### 3.4. Correlation between the Water Sorption and the Modulus of Elasticity

The E as a function of WS was plotted on a graph, and an exponential regression analysis was performed. The *t*-test with Bonferroni correction with regression analysis revealed that the specimens’ storage in water significantly influenced the modulus of elasticity (*p* < 0.05). The dry storage condition generated a high correlation coefficient between WS and E (R^2^ = 0.90, *p* = 0.004, Figure 4A). The higher the water sorption, the lower the modulus of elasticity of 1-SEAs (R^2^ = 0.85, *p* = 0.001, Figure 4B).

### 3.5. SEM/EDS Analysis

Representative SEM images of the 1-SEAs surfaces are presented in Figure 5. No significant changes were observed in the SEM images of the surfaces of the TD group and the HA group. The elemental mapping results are shown in Table 6. In the elemental analysis, more Ca was present on the HA grousp’s surface than on the TD group’s surface. (Figure 5).

## 4. Discussion

Hydrophilic monomers (i.e., HEMA) and water increase the hydrophilicity of the adhesives [15,16]. The HEMA content and the low pH of 1-SEAs may also negatively affect their polymerization [9], causing the elution of unpolymerized monomers, increasing the amount of water sorption by forming water immersion spaces [10], and jeopardizing the long-term prognosis of bonded restoration. Even after polymerization, those hydrophilic adhesives absorb water, leading to hydrolysis and plasticization of polymers, breaking the three-dimensional polymer chain networks, and reducing mechanical properties [17]. Thus, the hydrophilic and mechanical properties of polymerized 1-SEAs were evaluated in the current investigation. The null hypothesis was rejected, since using a hydroxyapatite plate surface to manipulate the 1-SEAs significantly increased their pH value, decreased WS and SL, and increased E (*p* < 0.05). 

When manipulated on the HA, the increase of 1-SEAs’ pH could probably be attributed to the hydroxide ions generated by the chemical reaction between the acidic functional monomers and hydroxyapatite [18]. 10-MDP, a standard content in all adhesives used in this study, is mildly acidic, and the reaction with hydroxyapatite is complex because the ionization proceeds in multiple stages [19]. CaHPO_4_·H_2_O is a sub-product of the reaction between 1-SEAs and hydroxyapatite, and it is also called dicalcium phosphate dihydrate (DPCD) or Burscheid (Appendix A). Since OH^-^ is not consumed in reaction formulas (D) and (E), OH^−^ remains a final product, and the pH value rises. It is suggested that the 1-SEAs’ pH change could be attributed to the reaction between 10-MDP and hydroxyapatite and the change in hydrogen ion concentration (Appendix A). In addition, the pH increase is expected since the human dentin presents a buffering capacity against acidic solutions [20].

All 1-SEAs significantly absorbed water, which agrees with other studies [7,21,22]. The resin polymer network absorbs water, cleaves the methacrylate ester bonds [23], softens the resin polymers, and diminishes the frictional forces between the polymer networks [12,24,25]. Thus, the water content affects the adhesives’ mechanical properties and the resin–dentin interface’s quality over time, possibly resulting in a reduction of dentine bond strength [26]. In addition, most contemporary 1-SEAs contain hydrophilic monomers, such as HEMA, which play an essential role in promoting the penetration of resin monomers into decalcified dentin and increasing water sorption [21] proportionally to the solvent’s concentrations [27]. Moreover, HEMA can form hydrogels before the monomers’ polymerization in an aqueous environment, forming a phase-separated poly-HEMA that is also prone to absorb water [12]. Consequently, the mechanical properties of the polymer network containing HEMA are jeopardized even after polymerization. 

Recent 1-SEAs removed or replaced this monomer with other hydrophilic monomers to address the shortcomings of the HEMA content. UBQ contains multifunctional hydrophilic amide monomers with more hydrophilicity after polymerization than HEMA [28]. Incorporating the amide monomers into 1-SEA may enhance the dentin sub-surface’s wetness, possibly benefitting the infiltration of hydrophobic resin monomers into demineralized dentin. However, according to the present data, the polymerized UBQ showed less water sorption than UBQ_exp_, which agrees with a previous study [29]; this low water sorption could be due to better polymerization [30]. Thus, the amide monomer in the UBQ composition is expected to generate a stable, long-lasting adhesive layer and a hybrid layer less susceptible to hydrolytic degradation, along with a better bonding performance than UBQ_exp_. 

Contrarily, WS values were significantly lower when adhesives were manipulated on HA than on TD. Most likely, the 10-MDP monomer in all three adhesives is responsible for the self-etching capability of the adhesive and chemically bonds to the hydroxyapatite [31]. The mildly acidic 1-SEAs dissolve only part of the hydroxyapatite around collagen fibers [15]. The remaining hydroxyapatite chemically bonds with the phosphate group of 10-MDP to form an insoluble calcium salt at the bonding interface. The results of SEM/EDS support this statement, showing that calcium could be detected on the 1-SEAs surfaces when the materials were manipulated in the hydroxyapatite plate (Figure 5 and Table 6). The calcium salts of 10-MDP are hardly dissolved in water or adsorbed on the hydroxyapatite surface. They may have increased the adhesives’ hydrophobicity due to the reduced number of polar groups incorporated into the materials.

Moreover, since the reaction with calcium could consume 10-MDP monomers, it is suggested that the number of 10-MDP monomers remaining in the bond layer decreases after the reaction with hydroxyapatite, contributing to the acidity reduction as observed for all the adhesives. In addition, since the solubility had an evident decrease when 1-SEAs were manipulated on HA, a lower elution of residual solvents and monomers is suggested. As a result, the insoluble calcium salts are unlikely to leach from the polymer matrix [12]. 

Only SBU showed a significant increase in flexural strength when the HA was used as a manipulation surface. However, all the materials presented significantly decreased flexural strength when the specimens were stored under wet conditions, regardless of the manipulation surface. In other words, the presence of water is crucial, weakening the strength of 1-SEAs. These results correlate well with earlier findings reporting that resin cement samples stored for 60 days in wet conditions presented lower flexural strength and modulus than those kept in dry conditions [32]. In the same study, significant changes in deflection at break were also registered [32]. Some possible explanations of the observed difference could be quality-influencing defects, such as the existence of porosities or a varying degree of polymerization.

E values decreased when specimens of all adhesives were manipulated on TD or HA and stored in water for 24 h. When a polymer is immersed in water, it is expected that the water bound to the polymer structure causes a plasticizing effect, leading to a decrease in the mechanical properties of the polymer [24,33]. Additionally, the absorbed water hydrolyzes the silane coupling agent used for fillers’ surface treatment, and delamination easily occurs at the interface between the filler and the resin matrix [34]. Furthermore, it can be assumed that water permeates the gaps caused by the silane removal, making it easier for the filler to separate from the organic matrix. This process progresses in a chain reaction, leading to the adhesives’ degradation.

Consequently, the contact area between the organic matrix and water increases when the adhesive surface becomes porous. Eventually, deterioration due to water absorption progresses, resulting in a decrease in E. UBQ_exp_ showed significantly lower E values than UBQ and SBU, which could be because of the higher amount of HEMA content in UBQ_exp_, resulting in more WS and lowered mechanical properties. Remarkably, the significantly higher E of UBQ than UBQ_exp_ could be attributed to the difference in the E between the amide polymer and poly-HEMA. Strong correlations between WS and E in dry and wet storage were found, following previous studies showing that a decrease in the mechanical properties of the adhesives correlated to their water sorption [7,33]. 

For all 1-SEAs, the adhesives manipulated on HA showed a higher E value than those manipulated on TD in dry and wet specimens. The increase of E values might have been caused by a decreased amount of polar functional monomer and decreased acidity of adhesives resulting from the reaction with the hydroxyapatite, which led to better polymerization and less water sorption. Demineralization by 10-MDP has been shown to create large amounts of calcium salts on enamel and dentin [35], so calcium salts should also be produced from the HA surface. It is speculated that parts of the two kinds of calcium salts (10-MDP-Ca salts and DCPD) could be precipitated in the adhesive due to the reaction between 10-MDP and hydroxyapatite, provoking improved mechanical strength. 

Although our results strongly suggest a beneficial interaction between 1-SEAs and hydroxyapatite, further research is required to investigate the materials’ behavior after reaction with enamel and dentin under simulated oral conditions.

## 5. Conclusions

Within the limitations of this study, it was concluded that contemporary one-step self-etch adhesives positively interacted with the hydroxyapatite plate surface, thus increasing their pH and decreasing water sorption and solubility. The manipulation of the adhesives on the hydroxyapatite plate also improved the adhesives’ flexural and elastic modulus in dry or wet conditions.

## Figures and Tables

**Figure 1 materials-15-08255-f001:**
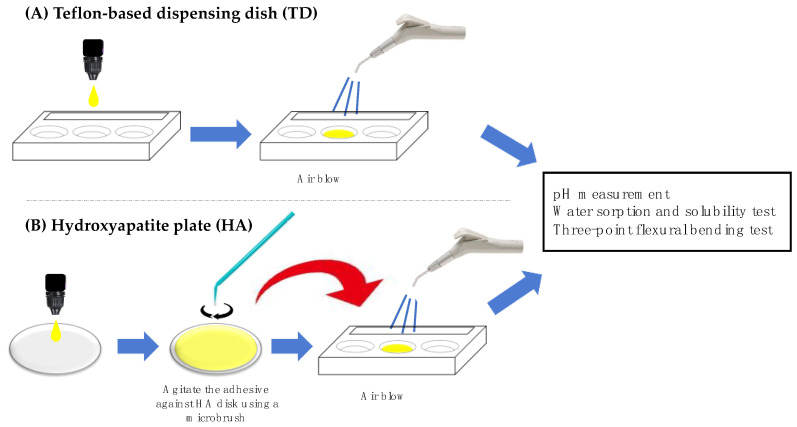
Schematic illustration of the manipulation and preparation of the 1-SEAs.

**Figure 2 materials-15-08255-f002:**
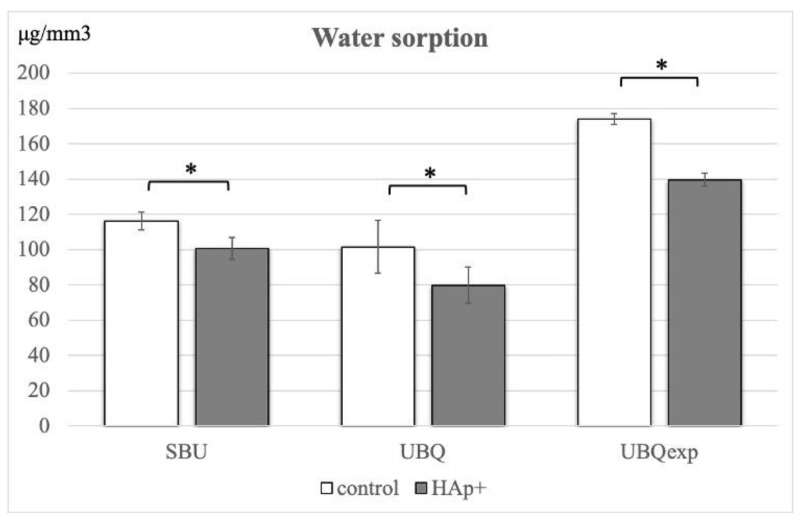
Water sorption values of the commercial 1-SEAs Scotch Bond Universal (SBU), Universal Bond Quick (UBQ), and Experimental 1-SEA (UBQ_exp_). Asterisks show statistically significant differences (*p* < 0.05) (*n* = 7).

**Figure 3 materials-15-08255-f003:**
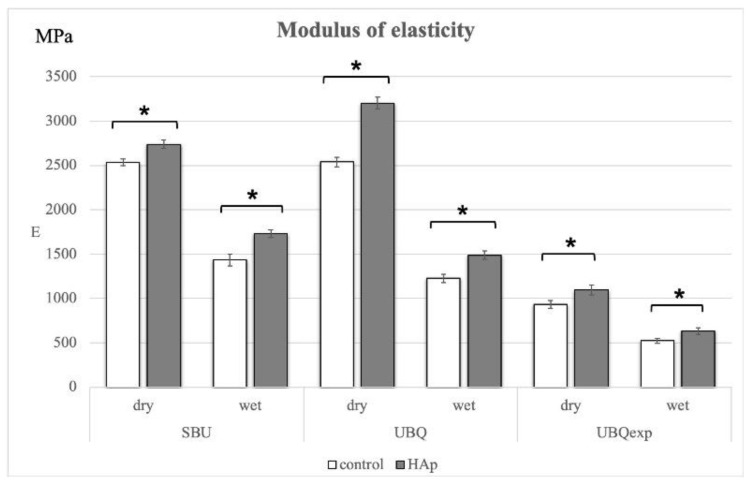
Values of the modulus of elasticity of the commercial 1-SEAs Scotch Bond Universal (SBU), Universal Bond Quick (UBQ), and Experimental 1-SEA (UBQ_exp_). Asterisks show significant differences (*p* < 0.05) (*n* = 8).

**Figure 4 materials-15-08255-f004:**
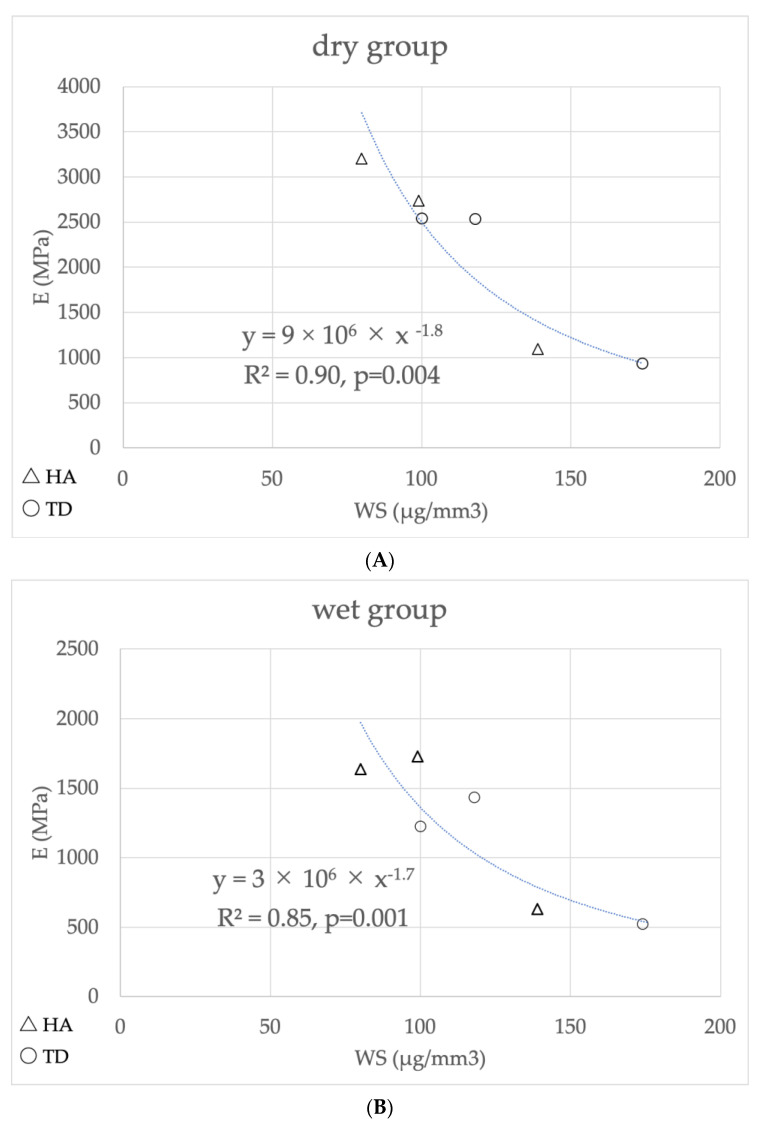
Regression analysis of the correlation between (**A**) E (dry group) and WS, (**B**) E (wet group) and WS.

**Figure 5 materials-15-08255-f005:**
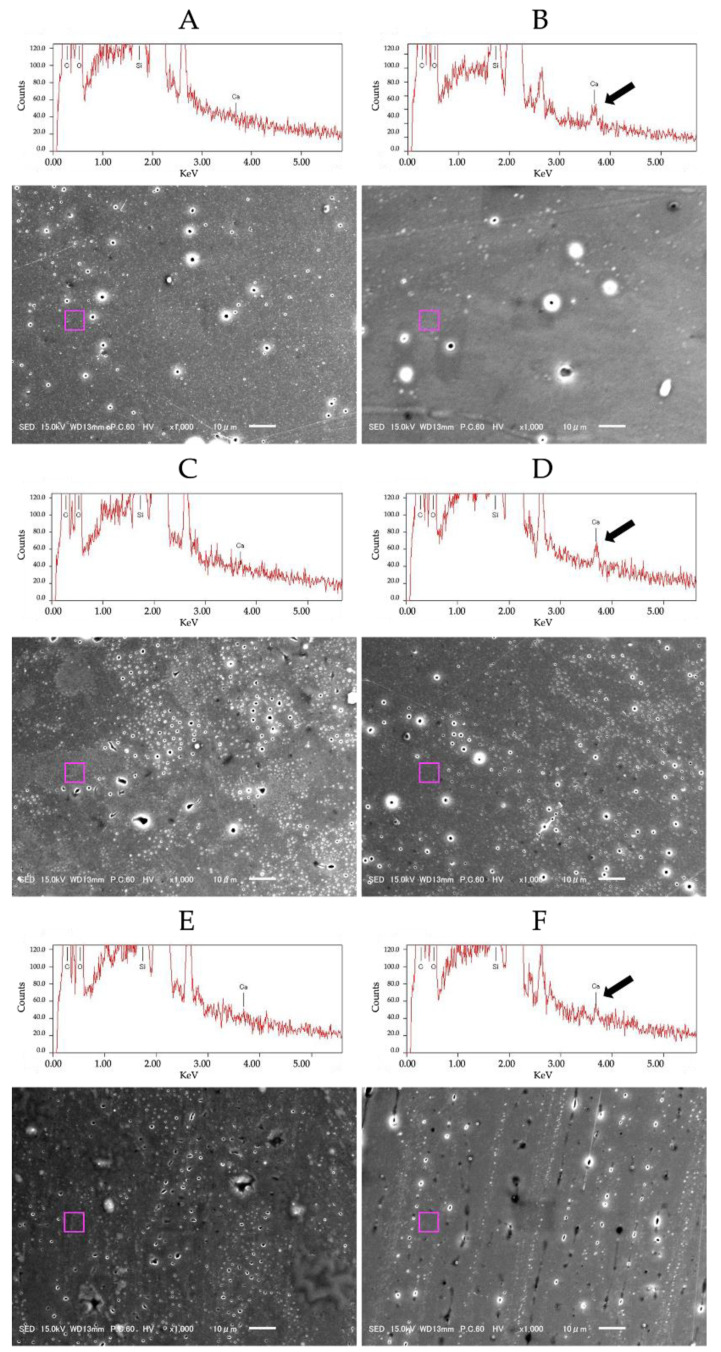
Representative SEM images of polymerized 1-SEAs specimens with related EDS spectrum. (**A**) SBU manipulated in TD, (**B**) SBU manipulated in HA, (**C**) UBQ manipulated in TD, (**D**) UBQ manipulated in HA, (**E**) UBQ_exp_ manipulated in TD, (**F**) UBQ_exp_ manipulated in HA. The pink squares indicate the location of the EDS analysis. The black arrow depicts the presence of calcium in the 1-SEAs manipulated in HA.

**Table 1 materials-15-08255-t001:** Composition of the one-step self-etch adhesives (1-SEAs) tested in this study.

Materials	Composition	Manufacturer
Scotch Bond Universal	10-MDP, HEMA, silane, dimethacrylate resins containing BisGMA, Dual Cure Activator, VitrebondTM copolymer, filler, ethanol, water, initiators	3M ESPE, USA
Clearfil Universal Bond Quick ER	10-MDP, BisGMA, HEMA, Hydrophilic amide monomers, Colloidal silica, Ethanol, dl-Camphorquinone, Accelerators, Silane coupling agent, Water, Sodium fluoride	Kuraray Noritake Dental Inc., Tokyo, Japan
Experimental 1-SEA	10-MDP, BisGMA, HEMA, Colloidal silica, Ethanol, dl-Camphorquinone, Accelerators, Silane coupling agent, Water, Sodium fluoride	Kuraray Noritake Dental Inc., Tokyo, Japan

10-MDP: 10-methacryloyloxydecyl dihydrogen phosphate; HEMA: (hydroxyethyl)methacrylate; BisGMA: bisphenol A-glycidyl methacrylate.

**Table 2 materials-15-08255-t002:** Results of the pH values of all three 1-SEAs.

	Adhesives’ Manipulation Surfaces	
	TD	HA
SBU	2.83 (0.06) ^Aa^	3.01 (0.10) ^Ab^
UBQ	2.42 (0.07) ^Ba^	2.55 (0.06) ^Bb^
UBQ_exp_	2.40 (0.03) ^Ba^	2.50 (0.04) ^Bb^

Different uppercase letters indicate comparisons between materials (rows), and different lowercase letters indicate comparisons between manipulation surfaces (columns) (*p* < 0.05).

**Table 3 materials-15-08255-t003:** Mean (SD) Water sorption [μg/mm^3^] and solubility [μg/mm^3^] values of 1-SEAs.

		Adhesives’ Manipulation Surfaces	
		TD	HA
SBU	WS	116.3 (3.2) ^Aa^	100.7 (3.7) ^Ab^
	SL	15.3 (3.6) ^Da^	13.2 (1.2) ^Da^
UBQ	WS	101.6 (5.1) ^Ba^	79.9 (6.3) ^Bb^
	SL	17.5 (2.5) ^Da^	7.8 (0.5) ^Eb^
UBQ_exp_	WS	174.1 (15.0) ^Ca^	139.5 (10.2) ^Cb^
	SL	117.5 (6.3) ^Ea^	85.9 (3.7) ^Fb^

Uppercase letters refer to comparisons between materials (rows), and lowercase letters refer to the comparisons between manipulation surfaces (columns) (*p* < 0.05).

**Table 4 materials-15-08255-t004:** Mean (SD) values (MPa) of the flexural strength (*n* = 8).

Adhesives	Storage Conditions	Adhesives’ Manipulation Surfaces	
		TD	HA
SBU	dry	148.9 (17.7) ^Aa^	189.7 (31.3) ^Ab^
	wet	73.9 (17.8) ^Ba^	101.1 (8.3) ^Bb^
UBQ	dry	160.5 (15.6) ^Aa^	173.1 (16.7) ^Aa^
	wet	68.2 (19.8) ^Ba^	76.4 (9.8) ^Ba^
UBQ_exp_	dry	59.6 (3.3) ^Ba^	69.5 (3.8) ^Ba^
	wet	28.0 (6.8) ^Ca^	33.2 (7.3) ^Ca^

The same superscript letters indicate no significant difference (*p* > 0.05). Lowercase letters refer to the comparison between manipulation surfaces (columns), and uppercase letters refer to the comparison between storage conditions within the same adhesive and the same manipulation surface (rows) (*p* < 0.05).

**Table 5 materials-15-08255-t005:** Mean (SD) values (MPa) of the modulus of elasticity (*n* = 8).

Adhesives	Storage Conditions	Adhesives’ Manipulation Surfaces	
		TD	HA
SBU	dry	2537 (37) ^Aa^	2736 (47) ^Ab^
	wet	1434 (67) ^Ba^	1729 (43) ^Bb^
	%ΔE	43.5	36.8
UBQ	dry	2539 (56) ^Aa^	3204 (64) ^Cb^
	wet	1226 (46) ^Ca^	1636 (46) ^Db^
	%ΔE	51.7	48.9
UBQ_exp_	dry	935 (44) ^Da^	1096 (54) ^Eb^
	wet	523 (27) ^Ea^	631 (37) ^Fb^
	%ΔE	44.0	42.4

The same superscript letters indicate no significant difference (*p* > 0.05). Lowercase letters refer to the comparison between manipulation surfaces (columns), and uppercase letters refer to the comparison between storage conditions within the same adhesive and the same manipulation surface (rows) (*p* < 0.05).

**Table 6 materials-15-08255-t006:** The elemental composition of each 1-SEA manipulated in the manufacturers’ Teflon-based dispensing dish (TD) or the hydroxyapatite plate (HA).

	Adhesives	Adhesives’ Manipulation Surfaces	Elements Formula				
			C	O	Si	Ca	Total
Atom (%)	SBU	TD	67.26	30.53	2.21	0.00	100.00
		HA	65.82	32.31	1.80	0.08	100.00
	UBQ	TD	69.25	28.23	2.52	0.00	100.00
		HA	69.27	28.25	2.39	0.10	100.00
	UBQ_exp_	TD	68.18	29.57	2.25	0.00	100.00
		HA	68.79	29.09	2.04	0.08	100.00
mass (%)	SBU	TD	59.47	35.96	4.56	0.00	100.00
		HA	58.09	37.98	3.70	0.23	100.00
	UBQ	TD	61.41	33.34	5.25	0.00	100.00
		HA	61.40	33.35	4.96	0.28	100.00
	UBQ_exp_	TD	60.42	34.91	4.67	0.00	100.00
		HA	61.11	34.43	4.23	0.23	100.00

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
