# Peer review of "Hydroxyapatite Affects the Physicochemical Properties of Contemporary One-Step Self-Etch Adhesives"

_materials, 2022, doi:10.3390/ma15228255_

Round 1
Reviewer 1 Report
The topic of the article seems to be interesting and up-to-date, however, some incorrectness was observed in the paper which needs to be corrected and the paper itself looks more like a research report than a scientific paper.
1. Lines 95, 99, 121, 127, 131, 141, 148, 209 - Numerous editorial errors were noted in the paper: should the numbers be in subscript? Check throughout the manuscript, including in the appendix;
2. Line 59 - incorrect designation of the standard - ISO 4049-2019 standard;
3. Lines 148, 163, 202, 203 - Editing errors - missing spaces - should be checked throughout the manuscript;
4. Line 162 - formula designation not in accordance with Journal Editorial requirements;
5. In my opinion the paper lacks a photograph showing the appearance of the materials examined;
6. Authors write: "The distribution of data and variance of data were analysed by the Shapiro-Wilk test and Levene's test. The variance of data was not equal among experimental groups." Why are the results of these analyses not included in the paper? Should be completed in the manuscript.
6. Conclusion needs to be developed
Author Response
Reviewer 1:
The topic of the article seems to be interesting and up-to-date, however, some incorrectness was observed in the paper which needs to be corrected and the paper itself looks more like a research report than a scientific paper.
Response: The authors appreciate Reviewer 1’s comments and suggestions to improve the manuscript. The modifications and corrections were highlighted in green in the attached file.
- Lines 95, 99, 121, 127, 131, 141, 148, 209 - Numerous editorial errors were noted in the paper: should the numbers be in subscript? Check throughout the manuscript, including in the appendix;
Response: The above-cited and other editorial errors were reviewed and corrected throughout the text. The corrections were highlighted in green in the attached file.
- Line 59 - incorrect designation of the standard - ISO 4049-2019 standard;
Response: The ISO designation was correct to the ISO 4049-2019 standard (pages 2 and 4).
- Lines 148, 163, 202, 203 - Editing errors - missing spaces - should be checked throughout the manuscript;
Response: The missing space errors were reviewed and corrected throughout the text. The corrections were highlighted in green in the attached file.
- Line 162 - formula designation not in accordance with Journal Editorial requirements;
Response: All the formulas and equations were corrected following the MDPI Journal Editorial instructions. The word “Equation” has been used.
- In my opinion the paper lacks a photograph showing the appearance of the materials examined;
Response: SEM images from the surface of each one-step self-etch adhesive manipulated in the manufacturers’ Teflon-based dispensing dish (TD) or the hydroxyapatite plate (HA) have been inserted. Please, refer to Figure 5 (pages 271-272).
- Authors write: "The distribution of data and variance of data were analysed by the Shapiro-Wilk test and Levene's test. The variance of data was not equal among experimental groups." Why are the results of these analyses not included in the paper? Should be completed in the manuscript.
Response: The results of those analyses (normality and homoscedasticity) were included on Page 5 (lines 187-190): “Data distributions indicated normality according to the Shapiro-Wilk test in all experimental groups (p>0.05). However, the data variance was not equal among the groups (p<0.05).”
- Conclusion needs to be developed
Response: The conclusion has been modified as follows: “Within the limitations of this study, it was concluded that contemporary one-step self-etch adhesives positively interacted with the hydroxyapatite plate surface, thus increasing their pH and decreasing water sorption and solubility. The manipulation of the adhesives on the hydroxyapatite plate also improved the adhesives’ flexural and elastic modulus, either in dry or wet conditions." (page 12).

Reviewer 2 Report
Thanks Author to choose Materials and MPDI to publish their manuscript
The study is well conducted, and the abstract and introduction allow readers to understand what the manuscript is about.
the materials and methods perfectly describe the procedures used.
The iconography is clear and quick to view and analyze.
A few typos are present such as (99) dark ... place?
Author Response
Thanks Author to choose Materials and MPDI to publish their manuscript
The study is well conducted, and the abstract and introduction allow readers to understand what the manuscript is about.
the materials and methods perfectly describe the procedures used.
The iconography is clear and quick to view and analyze.
A few typos are present such as (99) dark ... place?
Response: The authors thank Reviewer 2 for dedicating time to contribute to the refinement of the manuscript. The text has been checked for typos and other editorial errors. Modifications and corrections are highlighted in green throughout the manuscript.

Reviewer 3 Report
This paper evaluates the influence of the manipulation surfaces on the physical properties 21 of one-step self-etch adhesives. Two commercial and one experimental adhesive were investigated and subjected to water sorption (WS), solubility (SL), and flexural strength tests correlated to the type of substrate. This study is quite interesting and useful, but before publishing additional characterizations are needed, like the investigation of new chemical bonding in composite adhesives during manipulation and their microstructure. My suggestion is to perform FTIR spectroscopy and SEM analysis of manipulated adhesives; it will support a discussion of the chemical reactions during the process. Also, the clarity of Tables could be improved by putting units in columns.
Author Response
Reviewer 3:
This paper evaluates the influence of the manipulation surfaces on the physical properties 21 of one-step self-etch adhesives. Two commercial and one experimental adhesive were investigated and subjected to water sorption (WS), solubility (SL), and flexural strength tests correlated to the type of substrate. This study is quite interesting and useful, but before publishing additional characterizations are needed, like the investigation of new chemical bonding in composite adhesives during manipulation and their microstructure. My suggestion is to perform FTIR spectroscopy and SEM analysis of manipulated adhesives; it will support a discussion of the chemical reactions during the process. Also, the clarity of Tables could be improved by putting units in columns.
Response: The authors are grateful for Reviewer 3’s comments and suggestions, which have certainly contributed to the manuscript's improvement.
Following the reviewer’s suggestion, SEM/EDS analysis was performed. The sample preparation and the analytical conditions are described in the Material and Methods section, item 2.5., on page 5 (lines172-180). The results are shown on pages 9 and 10. Figure 5 and Table 6 were also inserted to illustrate the results.
Concerning Tables, the authors tried to put units in columns. However, it resulted in an overload of information inside the column title. Therefore, the authors chose to keep the units in each table’s title.

Round 2
Reviewer 1 Report
Thank you for the authors' responses to the review. I find all the responses satisfactory.
Reviewer 3 Report
The quality of the revised Manuscript is much improved with additional SEM-EDS characterization.
I recommended this paper for publication in the final revised form.